# Translation of Irrigation, Drainage, and Electrical Conductivity Data in a Soilless Culture System into Plant Growth Information for the Development of an Online Indicator Related to Plant Nutritional Aspects

**Tae In Ahn, Jung-Seok Yang, Soo Hyun Park** **, Heon Woo Moon and Ju Young Lee ***

Smart Farm Research Center, KIST Gangneung Institute of Natural Products, Gangneung 25451, Korea;
tiahn@kist.re.kr (T.I.A.); inenviron@kist.re.kr (J.-S.Y.); ecoloves@kist.re.kr (S.H.P.); 091689@kist.re.kr (H.W.M.)
* Correspondence: jyl7318@kist.re.kr; Tel.: +82-33-650-3703

**Abstract:** Electrical conductivity of the growing media or drainage indicates the nutritional conditions in the cultivation system. However, the nutrient uptake phenomenon has not been related well to the soilless culture system. Herein, we report on the design, theoretical analyses, and verification of a method for an online indicator related to plant nutritional aspects. Models for simulating nutrient and water transport in a porous medium were constructed for analyses of the nutrient uptake estimation method. In simulation analyses, we summarized the theoretical relationships between flow rates of total nutrients in a substrate and nutrient uptake. For concept validation, we conducted a greenhouse experiment for correlation analysis with the growth of tomato plants, conventional nutrient, and water management indicators, and developed online indicators related to plant nutritional aspects. Onsite application of the indicator showed a higher correlation with tomato yield than conventional management indicators, such as transpiration, irrigation, drainage ratio, leaching fraction, and electrical conductivity of drainage. In addition, to assess the usability of a nutrient uptake indicator as an onsite decision-making technique, data normalization was conducted. Through this, the time series responsiveness of a nutrient uptake indicator to the yield change was confirmed.

**Keywords:** nutrient uptake; decision support; water use efficiency; nutrient use efficiency

## 1. Introduction

Recently, agricultural production systems are undergoing a process of technological transition [1]. Furthermore, plant production platforms are also expanding to more advanced systems such as smart and vertical farms. One of the most observable changes is the rapidly increasing data flow [2]. Ultimately, this change could lead to the automation of decision-making. The derivation of useful information in the agricultural data chain is primarily based on the collection of data that reflects interactions between plants and the environment. Thus, appropriate interpretation of the sensor information under an automated data acquisition system of various plant production platforms is crucial for systematic linkage between plants and the cultivation system.

Technologies related to the capture, transfer, and storage of data are already being deployed in agricultural production systems [2,3]. In soilless culture, water management sensor-based research and development related to transpiration are actively being conducted. The data chain for plant water management with sensors is well established, such as those for measuring root zone moisture content [4–6], substrate weight [7,8], solar radiation [9], and humidity [10]. However, the translation of sensor data into knowledge of plants is still challenging [11]. Decision-making by farmers in commercial

farms is now being advanced to the level of a comprehensive analysis of water management with plant physiological statuses, such as vegetative and generative growth and root distribution [12].

Recent research is being extended to systemic linkages between cultivation and crops, which involves translating data from deployable sensors into plant state information, such as fresh weight and physiological condition of a plant [13–15]. However, transpiration is a phenomenon described primarily as a physical environmental condition in the plant–atmosphere continuum, although it reflects plant information, such as leaf area and stomatal conductivity [16–18]. On the other hand, the nutrient uptake phenomenon of plants is highly related to plant physiological statuses, such as relative growth rate, vegetative growth, generative growth, and plant stoichiometry [19–21]. Thus, expanding the data channels for plant physiological information, in addition to the transpiration data stream, could lead to the deployment of a more advanced decision support system. However, there are limited technological developments and research on an indicator related to plant nutritional aspects in data acquisition schemes within the soilless culture system.

A sensor associated with nutrient characteristics of the soilless culture system is the electrical conductivity (EC) sensor. However, to date, soilless culture systems have used EC data mainly for the management of the appropriate control status of nutrient concentrations in supplying the nutrient solution, substrates, and drainage. The drainage EC is a function of transpiration, nutrient absorption, and remaining available mineral nutrients [22,23]. Therefore, the EC of drainage can expect more information than just indicating nutrient concentration. In steady-state conditions of a system, the inputs and outputs of a component become the same as the internal process that produces or removes that component [24]. In practice, however, cultivation sites continue to experience nonhomogeneous distribution of nutrients in the root zone [23], intermittent water supply [25], and fluctuations in root zone water content and EC [9]. Some studies have provided estimates of individual nutrient absorption under soilless conditions [26,27], but no analysis has been conducted on the systems interpretation of the results, such as the effects of error factors or their utility. Thus, the systemic linkages between soilless culture system data such as irrigation, drainage, and EC to plant can have the potential to expand decision-making technologies in agricultural systems. However, little attempt has been made for direct corollary research to date to link EC data to the nutrient uptake characteristics and translates them into plant physiological information in the data acquisition system of soilless culture.

In this study, a nutrient uptake-related indicator of an arbitrary unit was extracted based on the supply and drainage volume of nutrient solution, and corresponding EC under simulated conditions of plant nutrient absorption, intermittent irrigation, and consequent EC variation and uneven nutrient distribution in a substrate. The sensitivity of the nutrient uptake-related index in an arbitrary unit was confirmed with comparisons of the change in nutrient uptake tendency under the simulated conditions affected by error triggering factors. Greenhouse experiments were conducted to investigate the correlations of the index calculated from the supply and drainage volume of the nutrient solution, and EC with the plant growth indicators to translate them into plant physiological information in the physical soilless culture system. In addition, a nutrient uptake-related indicator and the yield change data collected in this experiment were normalized to assess the potential usability as an onsite decision-support technique for yield-promoting nutrient and water management. Through the normalization, time-series responses of a nutrient uptake-related indicator were confirmed according to the relative change in yield between each treatment.

## 2. Materials and Methods

### 2.1. Simulation Analysis on Nutrient Uptake Estimation

Under ideal conditions such as a steady state, the inputs and outputs of a component in a system become the same as the internal process that produces or removes that component [24]. Thus, in theory, the steady-state condition of a media could provide an accurate response to the nutrient absorption by plants as the difference between the nutrients supplied to the media and discharged nutrients

from the media. However, in most soilless culture systems, nutrients and water are intermittently supplied by an automatic irrigation system, and in the case of a substrate, such as rockwool, conditions of nonuniform distribution of nutrients are formed in the root zone. In the present study, a soilless culture system model was constructed to analyze how the difference between nutrient supply and discharge follows changes in plant nutrient absorption under conditions that reflect these constraints.

The soilless culture system simulation roughly consisted of irrigation control based on the integrated solar radiation, changes in the water content of the substrate by irrigation and transpiration, and nutrient uptake (Figure 1a). The dynamic changes in incoming solar radiation were modeled by the total cloud cover model based on solar elevation [28,29]:

$$K^+ = K_0^+ \left(1 + b_1 N^{b_2}\right) \tag{1}$$

where $K^+$ is the reduced solar radiation by the total cloud cover; $K_0^+$ is the incoming solar radiation at ground level under clear skies, which is determined by solar elevation over seasonal time changes; $b_1$ and $b_2$ are the empirical coefficients; and $N$ is the total cloud cover. $N$ is a value between 0 and 1; closer to 0 corresponds to a clear day, and closer to 1 corresponds to a cloudy day. In simulation analysis, dynamic weather changes were simulated by moving $N$ between 0 to 1 in a random walk process. The irrigation of the soilless culture system was controlled based on the integrated value of solar radiation $K^+$ for simulating the general greenhouse irrigation automation method [9]. The transport of nutrients and water in a soilless culture system was simulated by the soilless culture system model of Ahn and Son [30] based on the nutrient transport model in a substrate condition [31,32]. For the absorption of nutrients, according to the concentration of nutrients in the substrate, the Michaelis–Menten equation was used. A nutrient absorption rate model incorporating the root surface area reflecting the absorption capacity of plants was used:

$$J^I = P_{RSA} \frac{J_{max}^I \left(C^I - C_{min}^I\right)}{K_m^I + \left(C^I - C_{min}^I\right)} \tag{2}$$

where $P_{RSA}$ is the root surface area (m$^2$), $J_{max}^I$ (mmol m$^{-2}$ min$^{-1}$) is the maximum absorption rate of nutrient I, $K_m^I$ (mM) is the Michaelis–Menten constant, and $C_{min}^I$ (mM) is the minimal concentration at which $J^I = 0$. The types of plant nutrients included in the simulation were K, Ca, Mg, NO$_3$, and P. In this simulation, a stochastic coefficient was applied to the nutrient absorption capacity of plants to detect changes in the rate of nutrient absorption under various conditions:

$$J^I = S_{cof} P_{RSA} \frac{J_{max}^I \left(C^I - C_{min}^I\right)}{K_m^I + \left(C^I - C_{min}^I\right)} \tag{3}$$

where $S_{cof}$ acts as a nutrient absorption factor and corresponds to a random walk process that increases or stops with a probability of $\lambda$ from the initial value of an absorption factor and decreases with a probability of $1 - \lambda$. For the transpiration model, the empirical version of the Penman–Monteith equation was used [33,34]:

$$Q_{trs} = a\left(1 - e^{-kP_{LAI}P_{VPD}}\right)K^+ + bP_{LAI}P_{VPD} \tag{4}$$

where $Q_{trs}$ is the transpiration rate (L min$^{-1}$), $a$ and $b$ are empirical coefficients, $k$ is the extinction coefficient in the plant canopy, $P_{LAI}$ is the leaf area index (LAI), and $P_{VPD}$ is the vapor pressure deficit (VPD). For the LAI used in the simulation, a fixed measured value was used. The leaf area of the tomato (*Solanum lycopersicum*) used in the LAI calculation was estimated by measuring the leaf area of the tomato in the cultivation experiment (measured at 2 January 2020). A non-destructive method was used for the leaf area estimation by measuring leaf width and length [35]. VPD was simulated to move in a random walk process between 0.5 and 2.0 kPa during simulation analysis to apply the stochastic fluctuation of transpiration in the simulation analysis. For simulation of the EC-based

nutrient solution supply method, the EC of the nutrient solution was calculated by converting the molar concentration of the nutrients into an equivalent concentration, and then the total equivalent concentration was converted to EC by the empirical conversion equation [36]. Calculation of the index related to nutrient absorption was conducted by summation of the difference between the nutrient inflow into the substrate and the nutrient outflow from the substrate:

$$\text{DNAI, Day Nutrient Absorption Index } = \sum_{i=1}^{n} (EC_i^{Sup} V_i^{Sup} - EC_i^{Drg} V_i^{Drg}) \tag{5}$$

where $EC_i^{Sup}$ and $V_i^{Sup}$ are daily EC and volume of irrigated nutrient solution, respectively, and $EC_i^{Drg}$ and $V_i^{Drg}$ are daily EC and volume of drained nutrient solution, respectively. Through simulation, various changes were made to the rate of nutrient absorption and drainage ratio of the soilless culture system, and the effect on the correlation between Day Nutrient Absorption Index (DNAI) and nutrient absorption was analyzed. Additionally, we compared the correlation between nutrient absorption and major indicators in nutrient and water management, such as irrigation amount, drainage ratio, leaching fraction, drainage EC, and transpiration. These indicators are available for direct data collection in the soilless culture system online and affect the growth of plants.

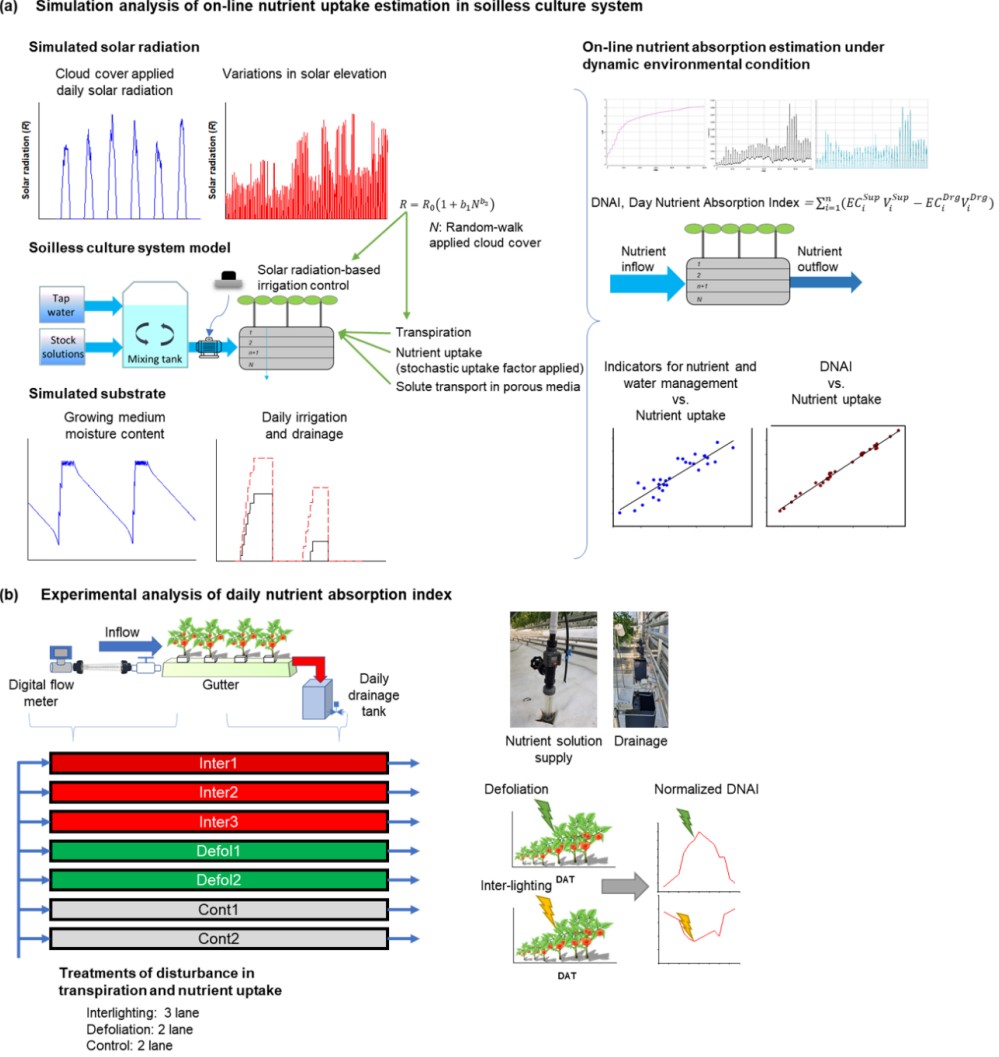

**Figure 1.** Schematic of simulation and experimental analysis of day nutrient absorption index. (**a**) Simulation analysis of on-line nutrient uptake estimation in soilless culture system. (**b**) Experimental analysis of daily nutrient absorption index (DNAI).

For the estimation and verification of the empirical coefficient of the transpiration model, water and solute transport in the substrate, the environmental data measured between 1 and 6 January 2020, at the experimental farm (37.8° N, 128.8° E) were used (data not shown). The change in the moisture content in the substrate because of the transpiration and supply of nutrient solution was verified by using the data from the substrate weight sensor (IReIS, RMFarm, Gangneung, Korea) at the same time as the environmental data were collected. In this simulation, the fresh weight of the plant and the dry weight of the rockwool substrate were not included. Therefore, the initial value of the substrate weight was corrected such that the initial value of the substrate weight and the initial weight of the substrate in the simulation were the same. RSA, which was used to reflect the total nutrient absorption rate, and the empirical coefficient of the transpiration model were estimated by a progress curve analysis that estimated the value that minimized the root mean square error (RMSE) between the measured and simulated values of the substrate weight change and drainage EC. Table 1 summarizes the main parameters used in this simulation.

**Table 1.** Parameters used for the simulations of the soilless culture system.

| Symbol | Description | Value | Reference |
|---|---|---|---|
| $P_{LAI}$ | Leaf area index | 7.4 | Measured in this study |
| $a$ | Transpiration empirical parameter | $1.52 \times 10^{-7}$ | Calibrated in this study |
| $b$ | Transpiration empirical parameter | $1.71 \times 10^{-4}$ | Calibrated in this study |
| $k$ | Extinction coefficient | 0.84 | [34] |
| $J_{max}^{K}$ | Maximum absorption rate | 0.009 | |
| $J_{max}^{Ca}$ | Maximum absorption rate | 0.003 | |
| $J_{max}^{NO_3}$ | Maximum absorption rate | 0.012 | |
| $J_{max}^{P}$ | Maximum absorption rate | 0.002 | [37] |
| $K_{m}^{K}$ | Michaelis-Menten constant | 3.185 | |
| $K_{m}^{Ca}$ | Michaelis-Menten constant | 0.617 | |
| $K_{m}^{Mg}$ | Michaelis-Menten constant | 0.252 | |
| $K_{m}^{NO_3}$ | Michaelis-Menten constant | 4.432 | |
| $K_{m}^{P}$ | Michaelis-Menten constant | 0.358 | |
| $C_{min}^{K}$ | Minimal concentration for uptake | 0.002 | |
| $C_{min}^{Ca}$ | Minimal concentration for uptake | 0.002 | |
| $C_{min}^{Mg}$ | Minimal concentration for uptake | 0.002 | |
| $C_{min}^{NO_3}$ | Minimal concentration for uptake | 0.002 | |
| $C_{min}^{P}$ | Minimal concentration for uptake | 0.002 | |
| $P_{RSA}$ | Root surface area | 0.8 | Calibrated in this study |

## 2.2. Experimental Demonstration of DNAI

Cultivation experiments were performed to confirm whether the predicted relationship between the DNAI and the absorption of nutrients by plants in the simulation was related to the growth index of plants under actual cultivation conditions. Cultivation experiments were conducted in a plastic

experimental greenhouse at the KIST Gangneung Institute of Natural Products (37.8° N, 128.8° E). Tomatoes (*Solanum lycopersicum* "Dafnis") were used as experimental plants and were cultivated on rockwool slabs (Grodan GT Master, Grodan, The Netherlands) placed in a hanging gutter of approximately 9.6 m in length with a planting density of 2.67 plants m$^{-2}$. An automatic drip irrigation system using an integrated solar radiation method was used for irrigation control. The cultivation area in the greenhouse was 384 m$^2$, consisting of a total of 18 hanging gutters, of which seven hanging gutters were used for DNAI measurement experiments (Figure 1b). DNAI was measured as described in Equation (5). The unit for measuring the input and output of nutrients was a hanging gutter. The volume of the daily supplied nutrient solution was measured by installing a digital flow meter (Water Smart Flow Meter, Gardena, Germany) in the pipe connected to each hanging gutter. The EC of the daily irrigated nutrient solution was measured after the completion of irrigation by placing one drop pin in a 2 L beaker. Daily EC of the drainage was measured after the daily irrigation was completed by placing a daily drainage collection tank (30 × 30 × 50 cm) equipped with an automatic discharge valve at the end of the hanging gutter. The volume of the drainage was measured by reading the water level in the daily drainage collection tank after completion of irrigation. Tomatoes used in the experiment were planted on 8 October 2019, and DNAI measurements were performed from 22 November 2019, 45 d after transplanting (DAT), to 9 April 2020, DAT 184. To compare the growth of plants with DNAI, the total yield of tomatoes planted in each hanging gutter was measured, and for the increase in stem length after the DNAI measurement, three plants per gutter were periodically measured.

### 2.3. Treatments for Disturbance Application on Nutrient Uptake and Transpiration

The relationship between DNAI and nutrient and water management indicators and plant growth is related to nutrient and transpiration. Therefore, in the cultivation experiment of this study, factors that could affect transpiration or absorption of nutrients were treated on the hanging gutter unit. By application of these disturbance factors, the performance of DNAI and its relationship with plant growth were analyzed. In this study, defoliation and inter-lighting were applied as disturbance factors that could affect transpiration or absorption of nutrients. In tomato cultivation, the effects of defoliation may increase or decrease the yield depending on the level of defoliation of leaves [38]. Additionally, an increase or decrease in leaf area also affects transpiration [39]. In the case of inter-lighting, it can affect the production of photosynthetic assimilation products of plants, which can increase yield [40]. Additionally, light is also linked to transpiration and acts as a stressor [17,41]. Therefore, inter-lighting or defoliation can act as a disturbance factor that can increase or decrease nutrient absorption or transpiration of plants. In this study, DNAI, drainage ratio, transpiration, drainage EC, irrigation amount, and leaching fraction were measured from DAT 45 without treatment application to each cultivation line.

Each treatment consisted of three lines of inter-lighting (Inter1–3), two lines of defoliation treatment (Defol1, 2), and two lines of control (Cont1, 2) for a total of seven hanging gutter lines. Inter-lighting treatment started on DAT 87. PPFD 168 μmol m$^{-2}$ s$^{-1}$ inter-lighting (LT080, Luco corp., Seoul, Korea) was used at a distance of 10 cm from the module and was placed in the central part of the plant canopy. Inter-lighting time was adjusted three times. The first inter-lighting operation time was based on the results of Tewolde et al. [40] in their tomato inter-lighting study, and a total of 12 h of operation time was applied from 22:00 to 10:00 the next day. However, after the initial inter-lighting treatment, apparent stress symptoms, such as leaf chlorosis and necrosis, were observed. Accordingly, the operation time was adjusted on the DAT 106, and the operating time of 5 h was applied from 17:30 to 22:30 per day. However, because the symptoms observed in the primary complement were neutralized, a total of 2 h of operation time was applied from DAT 115 from 17:30 to 19:30. Subsequent apparent growth was normally maintained, and no further time adjustments were performed. Defoliation treatment began on day DAT 132. In this study, all treatments, including the control, were subjected to conventional defoliation levels before treatment. In the conventional defoliation, 10–13 leaves counted from the top

5 cm and below were left. In the defoliation treatment, 3–4 more leaves were removed from the lower part than in the conventional defoliation.

To relatively compare the yield change between each treatment and the DNAI index, normalization of the measured values was performed, and the following equation was used,

$$x_{nor} = \frac{x - x_{min}}{x_{max} - x_{min}} \tag{6}$$

where $x_{nor}$ is the normalized value, $x$ is the DNAI to be normalized or the yield of each treatment, $x_{min}$ is the smallest DNAI of $x$ or yield per treatment, and $x_{max}$ is the largest DNAI of $x$ or yield per treatment.

## 3. Results

### 3.1. Simulation Analysis of DNAI

The results of the transpiration and irrigation simulations using the measured environment data in the experimental greenhouse as input data were compared with the change in substrate weight according to the transpiration, irrigation, and drainage in the substrate during the same period (Figure 2a). The change in weight measured by the substrate weight sensor tended to continuously decrease by VPD during the night when it was not irrigated, and the tendency was an increase in the rate of moisture reduction at sunrise before the first irrigation. Thereafter, as the transpiration by daily irrigation and solar radiation began, diurnal increase and decrease in water content in the substrate were observed, including a rapid decrease in moisture because of the end of irrigation before sunset and a change in weight of a constant slope after sunset were repeatedly observed. The simulation model was shown to follow the trend of variation in transpiration according to the change in light intensity and VPD in the daytime and change in VPD at night. Changes in the water content in the substrate were simulated by changing the moisture content in the substrate at the measurement of data and the RMSE 0.29 kg level. The EC of drainage, which changed according to the functional relationship of irrigation, nutrient absorption, and transpiration, was also simulated at the RMSE 0.66 dS m$^{-1}$ level (Figure 2b).

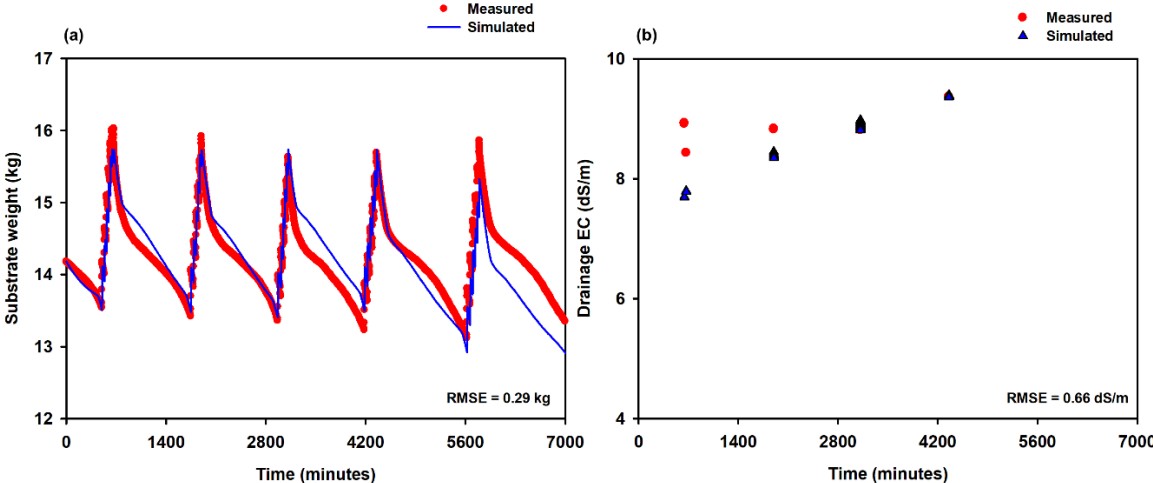

**Figure 2.** Comparison of simulated and measured substrate weight (**a**) and electrical conductivity (EC) of drainage (**b**) for verification of the soilless culture system model. No drainage occurred in simulation and experiment after 5600 min, and thus data of drainage EC for that period are also not presented.

To analyze the correlation between DNAI and nutrient absorption under conditions where stochastic changes in nutrient absorption and transpiration occurred, different solar irradiation conditions were confirmed for each simulation by applying a random walk process to the total cloud cover (Figure 3a). The solar radiation model showed the change in solar elevation with time and

location, and the solar radiation change accordingly, and an increase in the average solar radiation was observed over the month. Additionally, similar changes were observed in transpiration, and as VPD applied differently in each simulation, a certain range of variation was observed (Figure 3b). In the case of nutrient absorption, the absorption rate increased with the progress of the growth period of the plant but may decrease depending on the growth condition. Therefore, the stochastic change was applied to this, and different paths of the nutrient absorption factor change were determined for each simulation (Figure 3c,d). Two levels of the average nutrient absorption factor of 0.85 and 0.46 were applied to simulate changes in the nutrient absorption factor of various distributions between approximately 0.2 and 1.2.

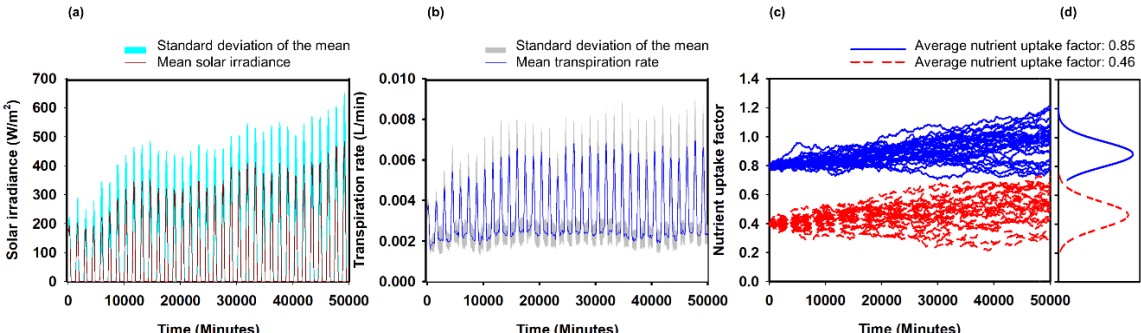

**Figure 3.** Random walk cloud cover and vapor pressure deficit (VPD) applied solar irradiance (**a**) and transpiration model (**b**), and stochastic changes in nutrient uptake factor (**c**) and its normal distribution (**d**).

Correlation analysis was performed between accumulated or average values of the transpiration amount, drainage ratio, irrigation amount, drainage EC, and leaching fraction, which were used as main indicators for the management of nutrients and water, DNAI, and absorption of nutrients (Figure 4). The coefficients of determination between the major indicators, excluding DNAI and cumulative nutrient uptake, were significant but extremely low negative correlations or positive relationships were observed. On the other hand, DNAI showed an extremely high correlation with cumulative nutrient absorption. However, in the distribution of the low nutrient absorption factor, a decrease in the coefficient of determination was observed compared to the high nutrient absorption factor (Figure 4f). Additionally, the changes in DNAI and $R^2$ values according to the drainage ratio were found to have a low correlation with the absorption of nutrients in the section with a low drainage ratio, and it was confirmed that the correlation increased with an increasing drainage ratio (Figure 4g). Additionally, the tendency for the $R^2$ value to decrease with a decrease in drainage ratio was greater in the low absorption magnification distribution.

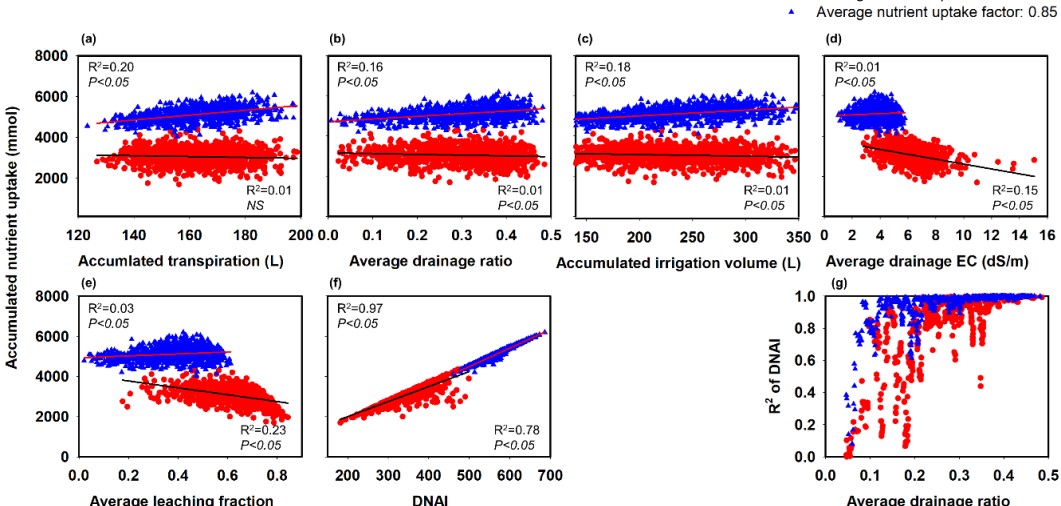

**Figure 4.** Comparison of correlation between simulated indicators in water and nutrient management, Day Nutrient Absorption Index (DNAI), and accumulated nutrient uptake under the different distributions of nutrient uptake factor (**a**–**f**); number of simulations: 1000. Changes in $R^2$ of DNAI according to average drainage ratio (**g**).

## 3.2. Correlation between DNAI and Plant Growth in the Cultivation Experiment

Through the measurement of daily nutrient irrigation amount, irrigation nutrient EC, daily drainage amount, and drainage EC during the DNAI measurement period, nutrient and water management indicators were collected for each hanging gutter (Figure 5). The accumulated transpiration increased relatively gradually but accelerated after DAT 100 (Figure 5a). The average drainage amount, the average drainage EC, and average leaching fraction were observed to be similar to each other (Figure 5b,c,f). These indicators were observed to increase rapidly after the initial decrease, and then decreased to 100 DAT, and then increased again. Depending on the condition of the nutrient solution pipe to each hanging gutter, a deviation in the water flow rate may have occurred. The difference between the most irrigated treatment and the least irrigated treatment in the final irrigation amount for each treatment was approximately 460 L. DNAI increased at the start of the measurements, but tended to decrease in all treatments around DAT 110 and increased until the end of the experiment (Figure 5e). The cumulative yield for each treatment increased with time, and no specific trend was observed on the graph (Figure 5g). However, a difference of approximately 11 kg was observed between the maximum yield and minimum yield. It was observed that the increase in shoot length increased at the start of the measurements, and then gradually became distinguished from each other among treatments after 80 DAT (Figure 5h).

The correlation between DNAI, nutrient management indicators, and cumulative yield was analyzed. During the entire period, a significant negative correlation was observed between the values for DNAI and the cumulative yield compared to other indicators (Figure 6b). Additionally, in the section where the daily drainage rate was low, DNAI was observed to have an extremely low $R^2$ value, and as the drainage amount increased to a normal level, a higher correlation was observed compared to that of other indicators (Figure 6a). Under conditions of low drainage, a positive correlation was observed, unlike other DNAI values, but did not appear as a significant correlation (Figure 6). Indices other than DNAI showed an extremely low correlation during most of the measurement period and were non-significant.

The correlation between the DNAI value, nutrient and water management indicators, and shoot length increase was analyzed. The correlation between DNAI and shoot length increase was mostly non-significant, except for the initial period of measurement and around DAT 115 (Figure 7). In these two cases, unlike the yield and its relationship, there was a positive correlation. Among the indicators for the management of nutrients other than DNAI, the average drainage EC showed the highest negative correlation with other indicators (Figure 7).

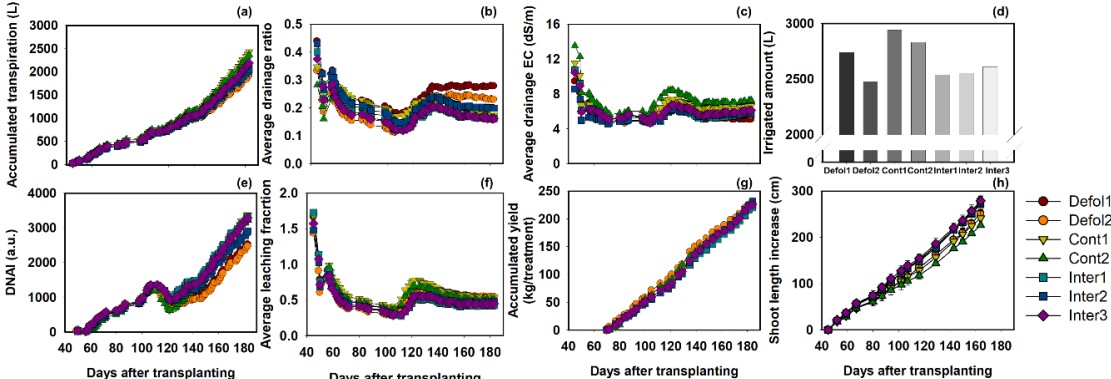

**Figure 5.** Accumulated transpiration (**a**), average drainage ratio (**b**), average drainage EC (**c**), irrigation amount (**d**), DNAI (**e**), average leaching fraction (**f**), accumulated fruit yield (**g**), and shoot length increase (**h**) during the period of the DNAI estimation experiment.

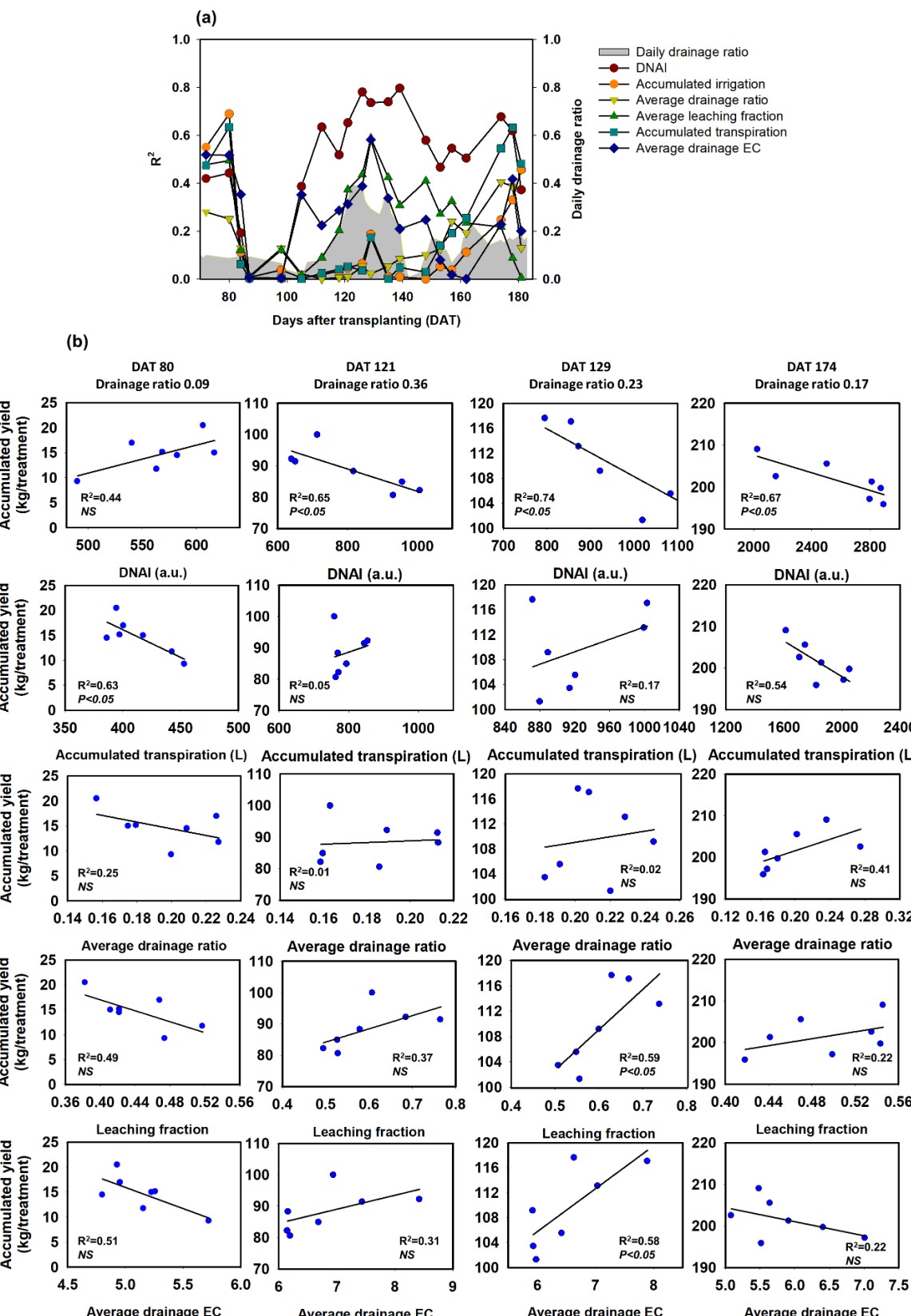

**Figure 6.** Changes in $R^2$ values between accumulated fruit yield and indicators in water and nutrient management, and DNAI during the period of the DNAI estimation experiment (**a**). Comparison of representative correlation between measured indicators in water and nutrient management, DNAI, and accumulated fruit yield (**b**).

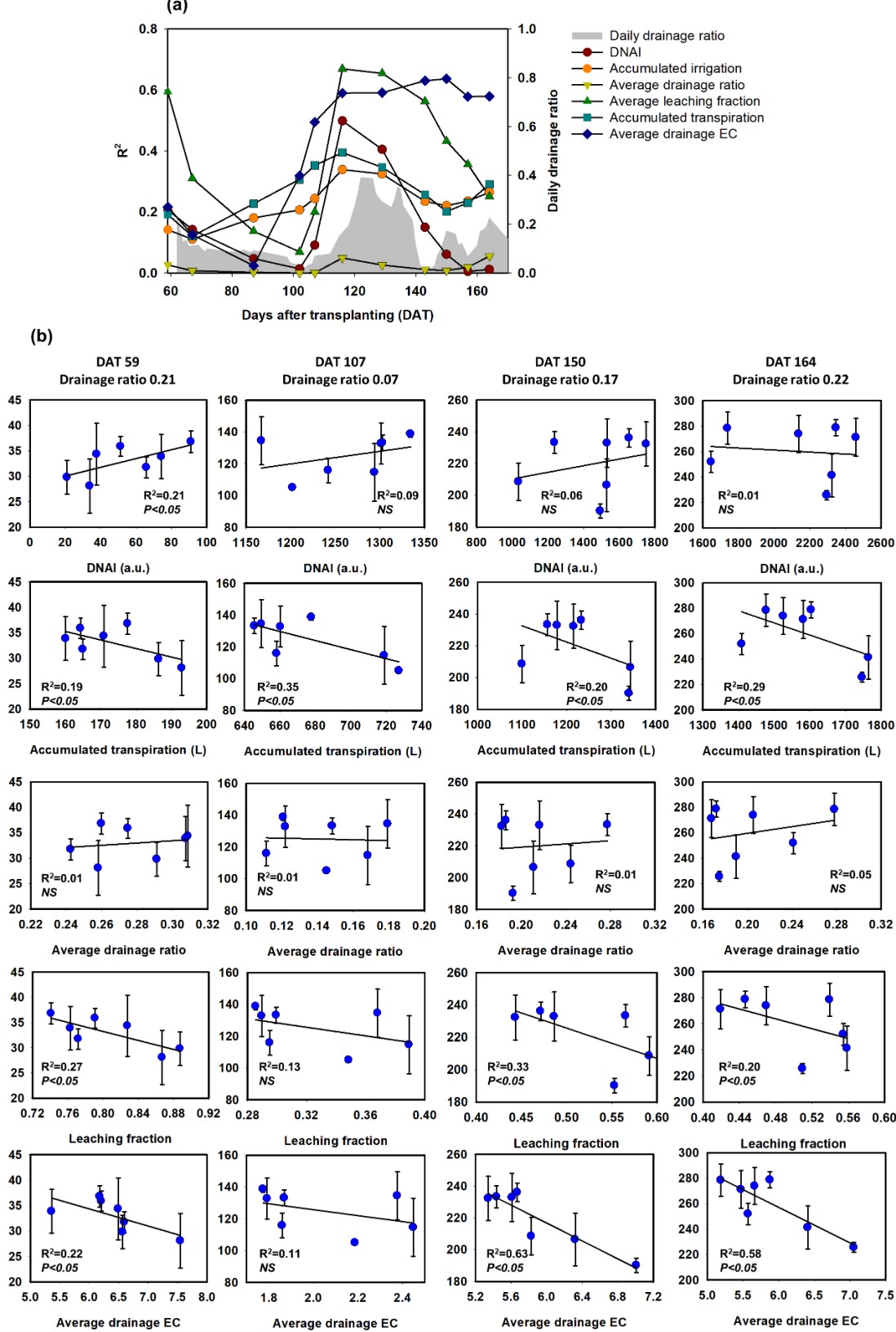

**Figure 7.** Changes in $R^2$ values between shoot length increase (mean ± SD) and indicators in water and nutrient management, and DNAI during the period of the DNAI estimation experiment (**a**). Comparison of representative correlation between measured indicators in water and nutrient management, DNAI, and shoot length increase (**b**).

### 3.3. Normalized Tomato Yield and DNAI in the Cultivation Experiment

The relative change in the yield and DNAI index between each treatment was observed using normalized values (Figure 8). In the case of inter-lighting treatment, a tendency of increase in DNAI was observed after the first 10 h inter-lighting treatment in Inter1 and 2 treatments. In the Inter3 treatment, the normalization value of DNAI was already close to 1, and no significant change in the normalization value was observed until the 3rd adjustment of the inter-lighting operation time. For the normalized value of the cumulative yield, a relative tendency of decrease in the cumulative yield was observed in the Inter1 and 3 treatments but not the Inter2 treatment, which was already close to 0 before treatment. In the secondary inter-lighting treatment, which was applied as stress symptoms, necrosis or chlorosis of leaves was observed, and an increase in DNAI was observed in the Inter1 and 2 treatments. After adjustment of the 3rd inter-lighting operation time, the tendency of decreasing DNAI was observed in the Inter2 and 3 treatments, and then the tendency of an increase in each yield normalization value was observed. However, the relative increase in yield was not observed in the Inter1 treatment even after the last adjustment.

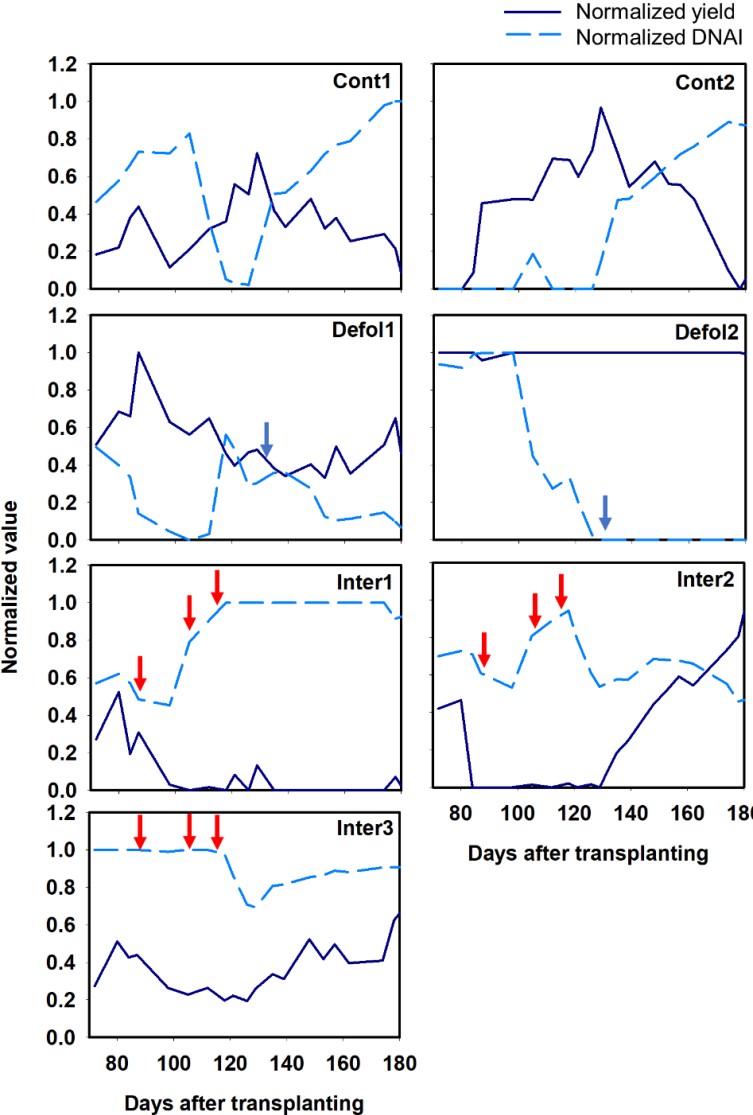

**Figure 8.** Changes in the normalized value of DNAI and accumulated fruit yield during the period of DNAI estimation experiment; blue arrows in defoliation treatment indicate the initial day of defoliation; red arrows in inter-lighting treatment indicate the initial day of inter-lighting and the subsequent adjustment of inter-lighting operation time.

Among the defoliation treatments, it was confirmed that the normalized value of the Defol1 yield led to a decreasing trend but changed to a slight increase after the defoliation treatment at 132 DAT. DNAI of the Defol1 treatment continued to decrease after the defoliation treatment. Defol2 treatment maintained the highest yield before treatment, and no change in the normalized value of the yield was observed even after the defoliation treatment. The DNAI of the Defol2 treatment was relatively high compared to that of other treatments, but then decreased rapidly and remained at the lowest level.

In Defol1, it was confirmed that the normalized value of the yield was changed to a slightly increasing trend after the defoliation treatment. The DNAI of Defol1 treatment showed a decreasing trend after the defoliation treatment. Defol2 treatment maintained the highest yield before treatment, and no change in the normalized value of the yield was observed even after the defoliation. The DNAI of the Defol2 treatment was relatively high compared to other treatments, but then decreased rapidly and remained at the lowest level.

In the control treatment, a tendency of increasing normalized values for yield was observed, and the tendency of DNAI to remain low or decrease was confirmed. However, from around DAT 130, the increasing trend for the yield shifted to a decreasing trend, and an increasing trend was observed for DNAI.

## 4. Discussion

In the study of the analysis of the simulation, the correlation between the cumulative nutrient absorption and the major indicators of nutrient and water management was significant; however, the coefficient of determination was extremely low (Figure 4a–e). These major indicators have not been modeled to have a direct functional relationship with nutrient absorption by plants. Thus, the low coefficient of determination can be seen originated from that the transpiration, irrigation, and drainage affected the change in the nutrient concentration of the root zone only. However, the models of plant transpiration based on the Penman–Monteith model included LAI as a representative parameter reflecting plant growth [33,42].

Additionally, the nutrient absorption model reflected the change in the absorption capacity of the plant because of the increase in the surface area of the roots [43]. In the more extended model, root growth was also interconnected with dry matter production of the shoot [37,44]. Therefore, a different trend could be predicted in an extended model in which all variables of the plant are interconnected.

However, the results from the simulation showed that DNAI could reflect, with a high probability, the change in the nutrient absorption trends in the root zone, even under conditions with disturbance factors, such as changes in plant transpiration and intermittent irrigation in the general cultivation system. The simulation analysis also showed that the higher the absorption rate of nutrients in the root zone, the higher the correlation observed (Figure 4f). The correlation was reduced under an extremely low drainage ratio, and the drainage ratio affected the correlation more under a lower nutrient absorption rate (Figure 4g). This appeared to be the result of a reduction in the corrective effect from the discharged nutrient amount in the DNAI calculations.

Regarding monitoring DNAI in the cultivation experiment, an increasing tendency was observed until DAT 110; however, a decreasing tendency occurred until 120 DAT (Figure 5e). In the case of the average leaching fraction corresponding to the ratio of cumulative discharge nutrients to cumulative supply nutrients, a decreasing tendency was observed until approximately DAT 110 (Figure 5f). This indicated that the discharged nutrients were low compared to the supplied nutrients. During this period, the daily drainage ratio was maintained at a low rate of less than approximately 10% (Figure 6a). However, DNAI decreased with an increasing drainage ratio after DAT 110, and a tendency in the correlation to increase between DNAI and cumulative yield after DAT 110 was observed. This result showed that DNAI was overestimated in the range with a low drainage ratio. As the drainage ratio increased, the trends in nutrient absorption by plants were reflected by the drainage, and this was applied as a correction effect. The effect of the drainage ratio on the correlation of DNAI in the experiment was consistent with the simulation's theoretical prediction.

After DAT 110, the $R^2$ value of DNAI showed a distinct change from the $R^2$ values of other indicators of nutrients and water management. To date, parameters such as transpiration, drainage ratio, leaching fraction, and irrigation amount had been widely used to improve nutrient and water management technologies. Regarding transpiration, it has been widely used in comparative studies of plant water stress and abnormal growth conditions [13,45,46]. However, transpiration under normal growth conditions is more dependent on the physical changes in the environment of the soil–plant–atmosphere continuum [16,17,42]. The drainage ratio is an indicator that can affect the growth of plants by adjusting the level of the drainage ratio [47,48]. However, the effect of the drainage ratio was fundamentally caused by an increase in heterogeneity of nutrient distribution and specific nutrient accumulation or deficiency in the root zone [23]. Leaching fraction—the rate of supply and discharge—played a role similar to that of the drainage ratio. Irrigation amount affects the changes in nutrient concentration and water content in a substrate, ultimately affecting plant growth [9].

In this study, the difference in the final irrigation amount was shown because of the deviations in the flow rate of each pipe line (Figure 5d). This can affect the drainage ratio and drainage EC in each treatment. However, the irrigation amount and yields were not highly correlated (Figure 6a). The effect of irrigation on the plants may vary depending on the difference between the defoliation status, light condition of each planting location, and microclimate conditions. In particular, the effect of the difference caused by a slight deviation in the flow rate of each pipeline can be too moderate to be detected as significant in the plant response.

In the case of absorption of nutrients from plants, nutrients are also stored in vacuoles in addition to the structure of the plants to maintain the ionic homeostasis of the plants [49]. However, nutrient accumulation is dominated by the stoichiometric growth of plants [19,50]. Therefore, the correlations between the yield and DNAI in the experiment, and nutrient absorption and DNAI in the simulation are considered to be the results reflecting the absorption of nutrients from plants.

In contrast to the relationship between yield and DNAI, there was no clear correlation with the tendency for shoot length growth (Figure 7a). However, when a significant correlation between DNAI and shoot length growth was found, a positive correlation was observed as opposed to a relationship with yield. An inverse relationship between shoot length and yield has already been reported [51], which can be considered to be in a trade-off relationship as the balance between nutritional and reproductive growth changes. The lower correlation with an increase in plant height compared to the yield vs. DNAI could be because the detection limit of the height increase effect was low. However, for the average EC of drainage, a significant negative correlation was observed with the increase in plant height (Figure 7b). The relationship between the increase in the root zone EC or the increase in the drainage EC has been reported in previous studies [52,53]. Although DNAI and plant height showed relatively low correlations, the correlation between drainage EC and plant height observed in this study seemed to have the potential to be utilized in further research on the technological sophistication of DNAI.

In this study, the effect of defoliation and inter-lighting treatment on DNAI and yield, which were applied to generate disturbance factors for transpiration and nutrient absorption, was qualitatively confirmed through normalized values (Figure 8). After the first inter-lighting of DAT 87, the normalized DNAI of Inter1 and −2 began to increase after a short period of decrease. At DAT 87, the normalized yield of Inter1 and −3 began to decrease. Prolonged light irradiation can act as a stressor on plants, and leaf chlorosis and necrosis can be observed as the symptoms [41,54]. In the case of the inter-lighting treatment, the first and second treatments acted on the apparent stress of the plant, as explained in the Materials and Method. Thus, the operation time of the inter-lighting was adjusted to 2 h after sunset in the third inter-lighting treatment. As a result, after the third adjustment of the inter-lighting operation time, decreases in normalized DNAI of the Inter2 and −3 treatments were observed. In addition, increases in the normalized yield of Inter2 and −3 were observed. However, in the Inter1 treatment, a decrease in normalized DNAI was not observed even after the third adjustment of the inter-lighting period, and the normalized yield did not respond.

In the case of the defoliation treatment, the relative decrease in the yield in Defol1 progressed from the beginning. However, after the defoliation treatment, a tendency of increase in the normalized yield and decrease in the normalized DNAI was observed. In the control treatment, an increase in normalized yield was observed before DAT 130. However, after approximately DAT 130, there was a tendency to decrease the yield again. Depending on the appropriate level of defoliation of tomatoes, it may lead to an increase in yield, and the appropriate level of defoliation may vary according to seasonal changes [38]. In the present study, the tendency of increase in normalized yield observed after defoliation in Defol1 and the decrease in normalized yield in Cont1 and 2 after a similar period may be related to the change in the appropriate leaf level.

This study may be limited in that it was not widely applied to other crops and other cropping seasons. The sensitivity of DNAI may be varied depending on the physicochemical property of the growing medium. Moreover, according to the crop's ability to absorb nutrients and water from the media, DNAI usability may vary. However, analyses of DNAI in this study suggested the possibility of integration of complex interactions between the conventional indicators in nutrient and water management on DNAI in the decision making of plant cultivation. In other words, adjustment of DNAI by manipulating cultivation management factors, such as nutrients and water input, defoliation, and supplemental lighting is expected to be applied to equalize the performance of agronomic manipulation, identify cultivation problems, optimize cultivation systems, and ultimately sustainable resource use. Furthermore, it is expected that DNAI approach can be expanded to other plant production systems such as vertical farms through further research for adjusting the time scale of data measurement and the sensor location.

## 5. Conclusions

The relationship between DNAI and the yield observed in this study indicated that DNAI can detect the effects of cultivation conditions, even with a relatively moderate difference in the conditions. The relative deviation in the flow rate of each pipeline in this experiment finally resulted in a difference in irrigation amount corresponding to a maximum of 410 L. Additionally, the difference corresponding to the cumulative yield of each gutter line up to approximately 11 kg was observed; however, a high correlation between irrigation and yield was not observed. This has implications for the complex influence of indicators (i.e., irrigation, transpiration, drainage ratio, and electrical conductivity) and DNAI has the potential to be used as an index that can be interpreted in a comprehensive method. As a result, this study confirmed that the DNAI could be associated with a high correlation to plant growth compared to conventional indicators in nutrient and water management by utilizing the data obtainable online in the cultivation system. Thus, DNAI showed potential usability as an onsite decision support technique for yield-promoting nutrient and water management. To develop DNAI as a decision-making technology, further studies to systematically link DNAI to nutrient and irrigation control are required. At the same time, verification in other crops also needs to be performed. We believe that further technological sophistication of DNAI will contribute to the efficient utilization of agricultural resources and automation of optimal water and nutrient management.

**Author Contributions:** Conceptualization, T.I.A., J.-S.Y., and J.Y.L.; investigation T.I.A. and H.W.M.; methodology, T.I.A. and J.-S.Y.; supervision, J.Y.L.; writing—original draft preparation, T.I.A.; writing—review and editing, J.-S.Y., S.H.P., and J.Y.L. All authors have read and agreed to the published version of the manuscript.

**Funding:** This work was supported by Korea Institute of Planning and Evaluation for Technology in Food, Agriculture and Forestry (IPET) through Smart Plant Farming Industry Technology Development Program, funded by Ministry of Agriculture, Food and Rural Affairs (MAFRA) (No. 119091-01-1-SB020 and 319029-01-1-SB020).

**Conflicts of Interest:** The authors declare no conflict of interest.

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
