# Peer review of "Translation of Irrigation, Drainage, and Electrical Conductivity Data in a Soilless Culture System into Plant Growth Information for the Development of an Online Indicator Related to Plant Nutritional Aspects"

_agronomy, doi:10.3390/agronomy10091306_

Round 1

Reviewer 1 Report

The research presented is interesting and timely.  Figures and graphs are hard to read and axes/legends could be enlarged.  This manuscript could be much more concisely written.  The words chosen do not clearly communicate the authors message to me.  

Author Response

Response to Reviewer 1 Comments

Thank you for your careful review of our paper. We found your comments and suggestions very helpful. Here are our detailed responses to the reviewer.

Point 1: Deletion of phrase for clarity (L31 in the initial submission manuscript).

Response 1: The sentence was modified as follows for clarity: “Ultimately, this change could lead to the automation of customary decision making by farmers.” → “Ultimately, this change could lead to the automation of decision making.” (L30-31 in the revised version of the manuscript).

Point 2: Relocation of “with sensors” in the sentence (L39 in the initial submission manuscript).

Response 2: According to the reviewer’s comment, we revised the sentence (L39 in the revised version of the manuscript).

Point 3: Reword for clarity (L44 in the initial submission manuscript).

Response 3: The sentence was modified as follows for clarity: “Recent research is being extended to this direction” → “Recent research is being extended to systemic linkages between cultivation and crops” (L44 in the revised version of the manuscript).

Point 4: High dimension (L45)?

Response 4: We deleted "high dimension" because “plant state information” includes similar context (L45 in the revised version of the manuscript).

Point 5: Activity (L46)?

Response 5: We changed “activity” in the sentence to “physiological condition of a plant” because we thought the expression of plant activity would be ambiguous (L46 in the revised version of the manuscript).

Point 6: Deletion of “root zone” (L47 in the initial submission manuscript).

Response 6: According to the reviewer’s comment, we deleted “root zone” in the sentence (L47 in the revised version of the manuscript).

Point 7: Addition of commas in the sentence (L51 in the initial submission manuscript).

Response 7: According to the reviewer’s comment, we added the commas in the sentence (L51-52 in the revised version of the manuscript).

Point 8: Deletion of “online” (L53 in the initial submission manuscript).

Response 8: According to the reviewer’s comment, we deleted “online” in the sentence (L54 in the revised version of the manuscript).

Point 9: Change “under” to “within” (L54 in the initial submission manuscript).

Response 9: According to the reviewer’s comment, we changed “under” to “within” in the sentence (L54 in the revised version of the manuscript).

Point 10: It remains a one-dimensional parameter in this discussion. I do not understand the reference to higher dimension (L59 in the initial submission manuscript).

Response 10: The change in EC of drainage is the result of transpiration and nutrient absorption by plants. Based on this, we have written it to mean that more information, other than merely monitoring total concentration, can be extracted from the EC change data of the drainage if the information on variables affecting drainage EC change is given together.  However, in order to clarify the delivery of this content, the following sentence was revised by reffering to the reviewer’s comments: “Therefore, the EC of drainage can expect higher dimensional information than just nutrient concentration. Under ideal conditions in which nutrients and water in the substrate are at a steady-state, input and output amounts of water and corresponding EC data provide highly correlated data with regard to plant nutrient uptake.” → “Therefore, the EC of drainage can expect more information than just indicating nutrient concentration. In steady-state conditions of a system, the inputs and outputs of a component become the same as the internal process that produces or removes that component [24].” (L59-62 in the revised version of the manuscript).

Point 11: Addition of “remaining available mineral” (L59 in the initial submission manuscript).

Response 11: In this sentence, “nutrients” refers to the nutrients remaining in the media. Thus, according to the reviewer’s comment, we added “remaining available mineral” in the sentence (L59 in the revised version of the manuscript).

Point 12: How, nutrient concentration would be more informative and correlate better with nutrition - furthermore, concentration is easily calculated into load when volume is known - allowing for calculations of mass for given mineral nutrient  (L60 in the initial submission manuscript).

Response 12: The answer to this comment is similar to Point 10’s. We wrote the response of this comment together in Response 10.

Point 13: Add reference  (L62 in the initial submission manuscript).

Response 13: We added reference based on the reviewer's comment. However, since the rerference corresponds to the general content of mass balance, the content of the text has been modified as follows: “Under ideal conditions in which nutrients and water in the substrate are at a steady-state, input and output amounts of water and corresponding EC data provide highly correlated data with regard to plant nutrient uptake.” → ” In steady-state conditions of a system, the inputs and outputs of a component become the same as the internal process that produces or removes that component.” (L60-62 in the revised version of the manuscript).

Point 14: We monitor EC to serve as a proxy for nutrient availability and subsequent uptake; therefore, I think this is misleading.  I do think you are correct to say there is little to no direct corollary research to date (L67 in the initial submission manuscript).

Response 14: We would like to adopt the reviewer’s suggestion in the manuscript when considering the functional roles of EC use in the soilless culture system. By referring to the reviewer’s comment, we modified the sentence as follows: “Attempts have not yet been made to link EC data to the nutrient uptake characteristics and translate them into plant physiological information in consideration of the data acquisition system in soilless culture systems.” → “Little attempt has been made for direct corollary research to date to link EC data to the nutrient uptake characteristics and translates them into plant physiological information in the data acquisition system of soilless culture.” (L68-71 in the revised version of the manuscript).

Point 15: “in” → “for” (L68 in the initial submission manuscript)?

Response 15: This sentence was written to indicate that there was little approach considering the data collection system of the soilless culture system. However, to clarify the content, the sentence was modified as follows by referring to the reviewer’s comment: “in consideration of the data acquisition system in soilless culture systems.” → “in the data acquisition system of soilless culture.” (L70-71 in the revised version of the manuscript).

Point 16: Addition of “the” in the sentence (L71 in the initial submission manuscript).

Response 16: According to the reviewer’s comment, we added the “the” in the sentence (L72 in the revised version of the manuscript).

Point 17: Change “actual” in the sentence to “physical” or “observed” (L78 in the initial submission manuscript).

Response 17: According to the reviewer’s comment, we changed “actual” to “physical” in the sentence (L80 in the revised version of the manuscript).

Point 18: Can you be more descriptive (L80 in the initial submission manuscript)?

Response 18: According to the reviewer’s comment, we modified the sentence as follows: “In addition, a nutrient uptake-related indicator and yield change data collected in this experiment were normalized to assess the usability as an onsite decision-making technique.” → “In addition, a nutrient uptake-related indicator and yield change data collected in this experiment were normalized to assess the potential usability as an onsite decision-support technique for yield-promoting nutrient and water management.” (L80-82 in the revised version of the manuscript).

Point 19: Delete or explain “line” (L80 in the initial submission manuscript).

Response 19: According to the reviewer’s comment, we deleted “line” in the sentence (L84).

Point 20: The reference is not suitable for such a claim.  The soilless system remains primarily a black box on nutrient status, which is only determined via destructive or displacement extracts unless referring to only hydroponics (L84-85 in the initial submission manuscript).

Response 20: According to the reviewer’s comment, we revised the sentence and the reference as follows: “In an ideal root zone condition in which the distribution of nutrients is entirely uniform, the nutrients absorption of plants can be easily calculated (Ref.).” → “Under ideal conditions such as a steady-state, the inputs and outputs of a component in a system become the same as the internal process that produces or removes that component (Ref.).” (L87-88 in the revised version of the manuscript). Reference: Van Noordwijk (1990) → Nordstrom (2007).

Point 21: Again, the system is never steady state under real-world conditions (L85-86 in the initial submission manuscript).

Response 21: By referring the reviewer’s comment, we revised the sentence to the following for theoretical assumption: “Under ideal conditions such as a steady-state, the difference between the nutrients supplied to the substrate and discharged nutrients from the substrate will accurately correspond to the nutrient absorption of plants.” → “Thus, in theory, the steady-state condition of a media could provide an accurate response to the nutrient absorption by plants as the difference between the nutrients supplied to the media and discharged nutrients from the media.” (L88-91 in the revised version of the manuscript).

Point 22: Change “with reference to the soilless culture system model of” to “based on” (L106 in the initial submission manuscript).

Response 22: According to the reviewer’s comment, we changed “with reference to the soilless culture system model of” to “based on” and modified the order of the references in the sentence as follows: “The transport of nutrients and water in a substrate was based on the nutrient transport model in a substrate condition [29,30] with reference to the soilless culture system model of Ahn and Son [31].” → “The transport of nutrients and water in a soilless culture system was simulated by the soilless culture system model of Ahn and Son [30] based on the nutrient transport model in a substrate condition [31,32].” (L109-111 in the revised version of the manuscript).

Point 23: Super wordy. State more concisely (L124-127 in the initial submission manuscript).

Response 23: According to the reviewer’s comment, we revised the sentence as follows: “ The leaf area of the tomato (Solanum lycopersicum) used in the LAI calculation was estimated by measuring the leaf area of the cultivated tomato at the same time as the environmental data used for simulation verification were measured.” → “The leaf area of the tomato (Solanum lycopersicum) used in the LAI calculation was estimated by measuring the leaf area of the cultivated tomato (measured at January 2, 2020).” (L128-130 in the revised version of the manuscript).

Point 24: Define online (L143 in the initial submission manuscript)?

Response 24: According to the reviewer’s comment, we revised the sentence as follows: “These indicators are available for online collection in the soilless culture system and affect the growth of plants according to the level of the indicators.” → “These indicators are available for direct data collection in the soilless culture system online and affect the growth of plants.” (L146-147 in the revised version of the manuscript).

Point 25: Repeated word (L144 in the initial submission manuscript)?

Response 25: By referring to the reviewer’s comment, we deleted “according to the level of the indicators” in the sentence (L146-147 in the revised version of the manuscript).

Point 26: Deletion of “for the verification” (L150 in the initial submission manuscript).

Response 26: According to the reviewer’s comment, we deleted “for the verification” in the sentence (L153 in the revised version of the manuscript).

Point 27: Deletion of “in the cultivation line” (L190 in the initial submission manuscript).

Response 27: According to the reviewer’s comment, we deleted “in the cultivation line” in the sentence (L193 in the revised version of the manuscript).

Point 28: Deletion of “for normalization conversion” (L219-220 in the initial submission manuscript).

Response 28: According to the reviewer’s comment, we deleted “for normalization conversion” in the sentence (L222 in the revised version of the manuscript).

Point 29: Addition of “diurnal” in the sentence (L233 in the initial submission manuscript).

Response 29: According to the reviewer’s comment, we added “diurnal” in the sentence (L235 in the revised version of the manuscript).

Point 30: Change “and” in the sentence to “including” (L234 in the initial submission manuscript).

Response 30: According to the reviewer’s comment, we changed “and” to “including” (L236 in the revised version of the manuscript).

Point 31: What does “according to the irrigation” mean (L236 in the initial submission manuscript)?

Response 31: When referring to the reviewer's comment, “according to the irrigation” was thought to be unnecessary, so we deleted it and modified the sentence as follows: “The simulation model was shown to follow the trend of variation in transpiration according to the irrigation in the daytime, change in light intensity and VPD, and change in VPD at night.” → “The simulation model was shown to follow the trend of variation in transpiration according to the change in light intensity and VPD in the daytime and change in VPD at night.” (L237-239 in the revised version of the manuscript).

Point 32: EC seems to be above normal values for a crop and would cause salt toxicity and impact water relations. Why no data at 5600 – 7000 (L243-244 in the initial submission manuscript).

Response 32: Although the EC shown in this study was observed in high level, it appears to be within the range of observable fluctuations in studies related to tomato soilless culture (Related study: Comparison of coconut coir, rockwool, and peat cultivations for tomato production: nutrient balance, plant growth and fruit quality, 2017; A targeted management of the nutrient solution in a soilless tomato crop according to plant needs, 2016).  

After 5600 min, enough drainage for EC measurement did not occur in the experiment and the simulation. Therefore, data during that period could not be presented. However, to avoid confusion, we added a description in the caption of Figure 2 as follows: “No drainage occurred in simulation and experiment after 5600 minutes, and thus, data of drainage EC for that period were also not presented.” (L246-248 in the revised version of the manuscript).

Point 33: Spell out (L344 in the initial submission manuscript).

Response 33: "Red leaf" is an error in the manuscript preparation process. Thank you for checking the error, and sorry for the confusion. We revised the sentence as follows: “It was confirmed that during the defoliation treatment that started on 132 DAT, the normalized value of the yield led to a decreasing trend and a slight increase after the red leaf treatment.” → “Among the defoliation treatments, it was confirmed that the normalized value of the Defol1 yield led to a decreasing trend but changed to a slight increase after the defoliation treatment at 132 DAT.” (L347-349 in the revised version of the manuscript).

Point 34: This is a very confusing and confounded sentence (L413-417 in the initial submission manuscript).

Response 34: According to the reviewer’s comment, we modified the sentence as follows: “Therefore, the correlations between the yield of fruit, which corresponds to the accumulating organ, and in the experiment, the correlation between DNAI and nutrient absorption in the simulation, and the relationship between the drainage ratio and DNAI observed in the experiment and simulation, are considered to be the results reflecting the absorption of nutrients from plants.” → “Therefore, the correlations between the yield and DNAI in the experiment, and nutrient absorption and DNAI in the simulation are considered to be the results reflecting the absorption of nutrients from plants.”(L419-421 in the revised version of the manuscript).

Point 35: Accumulating what? Please be specific with your words, adjectives and verbs (L414 in the initial submission manuscript).

Response 35: According to the reviewer’s comment, we modified the sentence as follows: “Therefore, the correlations between the yield of fruit, which corresponds to the accumulating organ, and in the experiment, the correlation between DNAI and nutrient absorption in the simulation, and the relationship between the drainage ratio and DNAI observed in the experiment and simulation, are considered to be the results reflecting the absorption of nutrients from plants.” → “Therefore, the correlations between the yield and DNAI in the experiment, and nutrient absorption and DNAI in the simulation are considered to be the results reflecting the absorption of nutrients from plants.”(L419-421 in the revised version of the manuscript).

Point 36: This conclusion seems far reaching without refererences (L428-429 in the initial submission manuscript).

Response 36: By referring to the reviwer’s comment on Point 36 and 37, we revised the sentence. The detailed information is presented in the Response 37.

.

Point 37: When looking at individual factors EC may have been significant; however, it poorly described the relationship adequately based on R2 values reported (L430-431 in the initial submission manuscript).

Response 37: We tried to address the implications for the utility of the drainage EC data in future technical sophistication research through this sentence. However, we think the sentence poorly deliver this content. By referring to the reviewer’s comment, we revised the sentence as follows: “In addition, the EC of the root zone is a factor influencing the physiological aspect of plants and is used as a major indicator of plant management in the actual field. The relationship between drainage EC and plant growth observed in this study suggested the possibility of developing a more integrated management index through linkage with DNAI.” → “Although DNAI and plant height showed relatively low correlations, the correlation between drainage EC and plant height observed in this study seemed to have the potential to be utilized in further research on the technological sophistication of DNAI.” (L431-434 in the revised version of the manuscript).

Point 38: Why not abbreviate here (L471 in the initial submission manuscript)?

Response 38: According to the reviewer’s comment, we used the abbreviation in the sentence (L468-484 in the revised version of the manuscript).

Point 39: Figures and graphs are hard to read and axes/legends could be enlarged.  This manuscript could be much more concisely written.  The words chosen do not clearly communicate the authors message to me. 

Response 39: According to the reviewer's comments, we have improved the visibility of the legends in figures and graphs. Also, the manuscript has been modified so that the content can be clearly communicated by referring to the reviewer's comments on the deliverability.

Reviewer 2 Report

General comments:

The manuscript entitled "Translation of irrigation, drainage, and electrical conductivity data in a soilless culture system into plant growth information for the development of an online indicator related to plant nutritional aspects" proposed an online nutrient update indicator based on supply and drainage volume of nutrient solution, and corresponding EC under different simulation conditions.   

The research article is well written and contributes to the existing knowledge. The experiments appear to be well planned, results are interesting, the ideas and methods are correct.

I suggest minor corrections to the authors, which are further listed:

Specific comments:

  1. The authors could include the impact of the research more clearly in the introduction section.
  2. The figures are informative but the font sizes of the texts inside the figures (1-7) should be increased for better presentation.
  3. Better to explain the reasons for being extremely low of the R2 values (Page-13; Line no: 362-363).
  4. The limitation of this research is simply mentioned (Page-15; Line no: 454-455), however, if applied to different crops, environments, and cropping seasons, what could be the major effects? Some details discussion regarding this issue would add value.
  5. Future plans and impacts of this research should be clearly and concisely mentioned at the end of the conclusion section.

Author Response

Response to Reviewer 2 Comments

Thank you for your careful review of our paper. We found your comments and suggestions very helpful. Here are our detailed responses to the reviewer.

Point 1: The authors could include the impact of the research more clearly in the introduction section.

Response 1: According to the reviewer's comment, the introduction has been partially revised to convey the impact of this study. Also, the following sentence has been added by referring to the comment: “Thus, the systemic linkages between soilless culture system data such as irrigation, drainage, and EC to plant can have the potential to expand decision-making technologies in agricultural systems. However, little attempt has been made for direct corollary research to date to link EC data to the nutrient uptake characteristics and translates them into plant physiological information in the data acquisition system of soilless culture.” (L66-71 in the revised version of the manuscript).

Point 2: The figures are informative but the font sizes of the texts inside the figures (1-7) should be increased for better presentation.

Response 2: According to the reviewer's comments, we have improved the visibility of the legends in figures and graphs.

Point 3: Better to explain the reasons for being extremely low of the R2 values (Page-13; Line no: 362-363).

Response 3: According to the reviewer’s comment, we added the explanation about the low R2 values as follows: “In the study of the analysis of the simulation, the correlation between the cumulative nutrient absorption and the major indicators of nutrient and water management was significant; however, the coefficient of determination was extremely low (Figure 4a-e). These major indicators have not been modeled to have a direct functional relationship with nutrient absorption by plants. Thus, the low coefficient of determination can be seen originated from that the transpiration, irrigation, and drainage affected the change in the nutrient concentration of the root zone only.” (L365-370 in the revised version of the manuscript).

Point 4: The limitation of this research is simply mentioned (Page-15; Line no: 454-455), however, if applied to different crops, environments, and cropping seasons, what could be the major effects? Some details discussion regarding this issue would add value.

Response 4: According to the reviewer’s suggestion, we added the content about the limitation as following: “This study may be limited in that it was not widely applied to other crops and other cropping seasons. The sensitivity of DNAI may be variated depending on the physicochemical property of the growing medium. Also, according to the crop's ability to absorb nutrients and water from the media, DNAI usability may vary.” (L457-460 in the revised version of the manuscript).

Point 5: Future plans and impacts of this research should be clearly and concisely mentioned at the end of the conclusion section.

Response 5: According to the reviewer’s comment, we revised the conclusion section as follows: “We believe that further technological advancement will contribute to the efficient utilization of agricultural resources and optimization of the cultivation system.” → “Thus, DNAI showed potential usability as an onsite decision-support technique for yield-promoting nutrient and water management. To develop DNAI as a decision-making technology, further studies to systematically link DNAI to nutrient and irrigation control are required. At the same time, verification in other crops also needs to be performed. We believe that further technological sophistication of DNAI will contribute to the efficient utilization of agricultural resources and automation of optimal water and nutrient management.” (L478-484 in the revised version of the manuscript).

Round 2

Reviewer 1 Report

All comments were addressed

Author Response

Dear Reviewer,

We are grateful for your careful reading of our text, and we also appreciate your suggestions for improving the manuscript.
